# Origins of Genetic Coding: Self-Guided Molecular Self-Organisation

**DOI:** 10.3390/e25091281

**Published:** 2023-08-31

**Authors:** Peter R. Wills

**Affiliations:** Department of Physics, University of Auckland, Auckland PB 92019, New Zealand; p.wills@auckland.ac.nz; Tel.: +64-9-923-8889

**Keywords:** mechanistic causation, computation, genetic coding, reflexivity, replication, translation, aminoacyl-tRNA synthetase

## Abstract

The origin of genetic coding is characterised as an event of cosmic significance in which quantum mechanical causation was transcended by constructive computation. Computational causation entered the physico-chemical processes of the pre-biotic world by the incidental satisfaction of a condition of reflexivity between polymer sequence information and system elements able to facilitate their own production through translation of that information. This event, which has previously been modelled in the dynamics of Gene–Replication–Translation systems, is properly described as a process of self-guided self-organisation. The spontaneous emergence of a primordial genetic code between two-letter alphabets of nucleotide triplets and amino acids is easily possible, starting with random peptide synthesis that is RNA-sequence-dependent. The evident self-organising mechanism is the simultaneous quasi-species bifurcation of the populations of information-carrying genes and enzymes with aminoacyl-tRNA synthetase-like activities. This mechanism allowed the code to evolve very rapidly to the ~20 amino acid limit apparent for the reflexive differentiation of amino acid properties using protein catalysts. The self-organisation of semantics in this domain of physical chemistry conferred on emergent molecular biology exquisite computational control over the nanoscopic events needed for its self-construction.

## 1. Introduction

According to the record of Socrates’ last dialogue [1], which he conducted immediately prior to ending his own life, the ancient philosopher expounded on the scientific way of thinking:
When I was a young man … I was an enthusiastic devotee of the branch of wisdom known as natural science. I thought it was a glorious achievement to know the causes of everything, why each thing comes into being, why it continues in being and why it perishes. I was always worrying myself over such questions as these: Does heat, by undergoing a sort of fermentation, give rise to animal life, as some people say?
However, he went on to deprecate attempts to comprehend everything in terms of matter and mechanism, concluding that such thinking displayed carelessness:
…[Anaxagoras] talked about such absurdities as air, ether and water … as if … in trying to state the causes of a particular action on my part, were to say:
‘He is now sitting on that couch for the following reasons. His body is composed of bones and sinews; the bones are hard and jointed, while the sinews are capable of contraction and relaxation and, together with the flesh and skin that contains them all, are laid around the bones; and therefore, as the bones are slung loose in their ligaments, the sinews, by relaxing and contracting, enable him to bend his limbs, and that is the cause of his sitting there with his legs bent’.
…It is utterly absurd to describe bones and such like as ‘causes’. True enough, I couldn’t have carried out what I considered right [accept the death sentence imposed by the City of Athens] unless I had bones, sinews, and the rest; but to say that these things are the cause of my doing what I do … would be an extremely slipshod way of thinking.
(The Graphical Abstract of this paper uses part of a faithful photographic reproduction of a two-dimensional, public domain work of art, Jacques-Louis David’s 1787 painting *The Death of Socrates*, taken from https://en.wikipedia.org/wiki/File:David_-_The_Death_of_Socrates.jpg, accessed on 1 July 2023.)

In this paper, I espouse the view that the advent of genetic coding was the life-defining moment in the chronicle of the local cosmos, a moment that ultimately allowed the causation of events in the material world to transcend the limitations imposed by physico-chemical mechanisms. Genetic coding subjugated the ineluctable laws of physics and their inexorable consequences to what manifests itself as a form of autonomous control or “guidance”, rendering those laws causally irrelevant to the main events in the realm of the living. Biology assigned physics to the realm of mere “substance effects”, practically enframing law-restricted matter as a resource for its further development. This historically contingent transition cannot be traced to the same origin as the transitions that separated the four fundamental forces from one another after the cosmological “Big Bang”. It was of a character completely different from those transitions and took place in a domain of activity governed not by quantum mechanics and general relativity but by the formation and processing of referential information. In that domain, chemical groups took on an immaterial, abstract status, that of digital symbols, a transition that was made possible by *reflexivity*, a pattern of self-referential information that arose as a side effect of an autocatalytic chemical cycle: the information in some extant patterns of symbols was processed computationally to produce the molecular machinery needed to execute the steps of that very computation. (The term *computation* is used in this paper to refer broadly to *information processing*, corresponding to the stepwise transformation of patterns and defined elements thereof embedded in physical reality. This usage is not intended to imply rigid conformity to the theory of automata, formal languages, recursive functions, decidability, Turing machines, logic gates, computational complexity, etc., or that computation need be strictly algorithmic.)

From that point on, the constructive information processing of biology rapidly expanded until its effects dominated the surface of the planet, ordering and transforming the molecular composition of the entire geosphere, eventually producing in the universe a rare island of what passes for “intelligence” in the parlance of contemporary society, including its scientific literature.

I begin by contrasting this perspective with more conventional answers to the question, “What is life?” and corresponding explanations of its origins. That leads to an enquiry into the role of self-organisation in biological systems and their origins, especially the essential molecular biological processes of replication and translation. The centrality of information processing in biology begs a discussion of the character of “guidance” and its relationship to agency, casting functioning biological systems as instances of *self*-guided self-organisation. Analysis of model Gene–Replication–Translation systems [2] serves to illustrate how polymer sequence information accumulating in macromolecular quasi-species [3,4] can guide the self-organisation of genetic coding, which in turn can guide Darwinian selection within the population of the polymeric genes. The underlying mechanism of quasi-species bifurcation enables a massively parallel evolutionary search that quickly finds new levels of chemical reaction specificity and information-processing accuracy, progressing from one level to another through saltations that each generate new possibilities of functional, structural and computational complexity.

## 2. Differing Conceptions of Biology’s Foundations


*Thesis: Molecular biology encompasses both chemical and computational modes of description and analysis but lacks a theory of the natural unification of matter and information processing that adequately identifies the essence of Life.*


Biologists commonly use two complementary modes of thinking and reasoning about causation. The first mode is based on the assumption that anything in the material world can be understood by dividing it into separable parts, studying the parts individually and discovering how they join together, interact and react with one another. At the basic level of molecules, this is the mode of thinking of biochemistry. For example, protein synthesis is understood in terms of the energy-driven process whereby a ribosome moves along a messenger RNA as it sequentially binds aminoacyl-tRNA molecules, the amino acid moieties of which are transferred in order, one by one, to the C-terminal end of the growing peptide chain. The molecular mechanics of the process are very complicated, but after decades of intensive research, we know the detailed atomic structure of each of the more than one hundred proteins and RNA macromolecules involved in the operation and how each contributes to the functioning of the protein production line. 

The second mode of thinking is that of molecular biology, which, in relation to protein synthesis, envisages the application of a code to translate information stored in nucleic acid sequences into the amino acid alphabet of protein sequences. The code is used to choose each amino acid in turn so that the resulting protein, when folded, serves its proper cellular function. Within this mode of thinking, ribosomes and their cofactors constitute nanoscopic dual-tape Turing machines that operate computationally to control protein production [5]. These machines take messages encoded in the form of nucleotide sequences as input and, applying the genetic code, produce corresponding strings of amino acids as output. The code is manifest in the substrate specificity of the cellular population of aminoacyl-tRNA synthetase (aaRS) enzymes, soluble cofactor components of the Turing machine that effectively control what is presented to the ribosomal write-head at the peptidyl transferase centre, while the relatively distant read-head at the interface of the ribosomal subunits ratchets its way along the mRNA (Figure 1).

Biologists swap seamlessly between these two modes of thinking, even in the course of conducting a single procedure, e.g., to produce and characterise a protein. The first step is usually the synthesis of a gene whose sequence encodes a protein with the desired amino acid sequence. This gene is spliced into a plasmid and introduced into a bacterial culture, causing the cells to manufacture an enormous number of copies of the protein—for example, 1 mg of a 60 kDa protein is ~10^16^ molecules—before the cells are lysed and the protein extracted. Standard biochemical procedures of molecular separation and analysis, for which the protein is “just a conglomeration of atoms”, are employed to isolate and characterise the product, although its amino acid sequence may be determined to ensure that it is the exact translation product of the synthetic gene. 

The computational perspective of molecular biology was first suggested by Schrödinger [6], who talked about chromosomal (genetic) material containing a *codescript* and possessing the “executive power” necessary to give effect to such a programme. He knew nothing of information theory [7] or the structure of DNA [8], but surmised that cells could not exist without internal reference to some decipherable pattern of atoms (an “aperiodic crystal”) stable enough to remain “unperturbed by the disordering effect of the heat motion for centuries”. He likewise insisted that the ordered configuration of molecular biological processes could only be maintained at the expense of continual entropy production in the surroundings. For Schrödinger [6], the existence of a quasi-symbolic codescript and molecular organisation dependent on external thermodynamic dissipation gave the best contemporary scientific answer to the question “What is Life?”, but it was left to Prigogine [9] to recognise the necessity of chemical systems crossing some threshold of dissipation before they could attain forms of spatio-temporal self-organisation reminiscent of biological systems.

The more recent Joyce-NASA definition of life as “a self-sustaining system capable of Darwinian evolution” [10] implicitly assumes the intimate connection between information and matter described by Schrödinger [6] without providing further illumination. It strangely ignores what has since been learned about genetic coding, the advent of which is reasonably interpreted as having framed the genotype-phenotype relationship that made whole-system Darwinian selection possible. The definition’s unqualified appeal to ‘Darwinian evolution’ obscures the central biological problem of explaining the autonomy and spatio-temporal persistence of complex, individual organisms that have to overcome the brutality of the second law of thermodynamics prior to having any chance of participating in the Darwinian game of survival. Biological subsystems can be regarded as self-sustaining only as a result of their astounding functional organisation, something which is not derived from their molecular composition per se. From a thermodynamic perspective, the all-important functional organisation is definitely not self-sustaining. It requires, as Schrödinger [6] already emphasised, an environmental source of free energy that can be exploited to drive all of the system’s internal processes. Overall, the biosphere drives cycles of chemical transfer by continually degrading free-energy-rich sunlight to lower-temperature thermal radiation, which is lost from the earth’s surface into the darkness of space. Without some such one-way transfer of free energy to sustain them, no biological systems could exist.

Likewise, the epithetic formula of Küppers [11] and Davies [12], “Life = Matter + Information”, conveys nothing concerning the cosmic significance of the co-dependent relationship between living matter and biological information. In their unpacking of the formula, these authors acknowledge that the semantic dimension of biological information is indispensable to its character, but the supremacy of physico-chemical causation is maintained. As stated by Küppers [13], “living matter is … governed … *only* by the physical and chemical properties of biological molecules”, and “[the] highly coordinated interplay [between genetic instructions and chemical functionality] is based *exclusively* on the known laws of physics and chemistry” (*emphasis added*). However, information was not, as the formula implies, a simple “add-on” that transformed the matter on the surface of the Earth into a cauldron of orchestrated nanoscopic computation. Rather, if we extend Küppers’ metaphor of nucleic acids and proteins as “the legislative and executive sides of government”, or as Schrödinger [6] put it, “law-code and executive power—or … architect’s plan and builder’s craft”, the origin of life was a takeover, a coup, whereby information, availing itself of vulnerabilities in the simplicity of matter’s orderly structure and behaviour, took control of it by imposing its own meaning on it. Clearly, we should be wary of describing natural processes as they are perceived through the lens of various politico-historical processes endemic to our species’ behaviour, but we should remain attentive to the implicit core of such thinking, the attribution of agency to information. 

Biologists avail themselves of the reality of computational causation in molecular biological events whenever they engineer cells by introducing into them genes with sequences that they have devised, whether the choice of sequence has been made with the bulk production of a protein in mind or the breeding of animals with novel characteristics. Computational thinking complements physico-chemical thinking in the modern practice of biology, in which case it behoves biologists to ask how the coincidence of arbitrary sequence patterns and molecules that selectively recognised them formed the kernel of the first self-organised information-processing systems, from which all life has evolved. This theme has long been embraced by workers such as Carter [14,15] and Kaçar [16], who have also conducted extensive experimental work, the biological interpretation of which would be inadequate outside of the assumption that molecular biology is fundamentally computational. 

Before the origin of life, information in the form of arbitrary molecular configurations, such as the sequences of heteropolymers, had no coherent means of participating in the constraint and guidance of chemical processes, especially with the attendant specificity required for the maintenance of self-reproducing molecular machines [17]. What then were the crucial events whereby molecular information and chemical processes became integrated into a self-organised, dissipative structure, in which an array of molecular components cooperatively carried out all of the specific steps required for their own production as a computational interpretation of the information? Organisms exist only by processing genetic information, that is, through computation, and the required computation is possible only on account of it producing the molecular components that carry it out, an essential feature of the abstract self-reproducing machines (automata) of von Neumann [18]. This brings us to the concept of reflexivity, with which we seek to encompass the strange aspect of closure that characterises biological causation, a logical loop that crosses levels and, by some apparent trick of topology, returns to where it started [19].

## 3. Reflexivity: Replication and Translation


*Thesis: Both core processes of molecular biological computation, the replication and translation of genetic information, are examples of self-organising autocatalysis. Translation represents a mapping from molecular sequence information held in genes onto the catalytic functions of protein catalysts. The mapping is reflexive, operated by the physically constructive action of its own products. Translation is a self-organising, constructive, computational process.*


The process of molecular replication foreshadows, rather dimly, the computational causal closure of biological processes that genetic coding enabled. Some molecule X—typically a linear heteropolymer made up of monomers chosen from an “alphabet” of *λ* monomers, such as nucleic acids made up of concatenated nucleotide monomers chosen from the quarternary (*λ* = 4) alphabet {A, C, G, T/U}—acts as a template for its copying: the extant copy of X intervenes in the process of the progressive monomer condensation reaction, ensuring that the sequence of the product is the same as that of the template. (We ignore for now that Watson–Crick base-pairing dictates that this is in fact a two-step process, with the complementary sequence as an intermediate template.) Likewise, X could be a low molecular weight organic compound able to catalyse its own formation from two or more smaller reactant components. A good biochemical example of autocatalysis is the formose reaction, in which the input of one molecule of glycolaldehyde produces two [20], and a simple chemical example is the decomposition of arsine (AsH_3_) to form arsenic (As), which is a catalyst of the decomposition process. In either case, molecules of X have separate roles, appearing as both the product and a catalyst of the overall chemical process, which is represented in three different ways in Figure 2.

The third representation in Figure 2, which could be understood as indicating nothing more than dynamic feedback in the synthesis of the X, signifies that the process is reflexive within the context of distinguishable molecular roles (reactant, product, catalyst), the definitions of which are arbitrary in respect of any molecule’s intrinsic properties that arise from its own quantum mechanical structure. Thus, the reflexive closure of chemical processes entailed in autocatalysis is something that arises contextually and relates to patterns in the description of a system rather than it being a physico-chemical property of the system’s constituent molecules.

In the case of a replicating polymer, the supra-physicochemical pattern of interest is the polymeric sequence, which potentially carries information. The symbol-by-symbol copying of polymer sequences is never exact, resulting in a distribution of replication products, even from a single parent molecule. The best that can be expected is that the consensus symbol occupation at any position in the product sequences is the same as that of the parent sequence, but the reality of replication can diverge much further than the norm of “copying” to produce a stably reproducing quasi-species in which the overall symbol occupancy at any position is not far from random [3]. Therefore, the symbol X in Figure 2 actually represents the *λ^ν^* individual sequences that are potential members of a quasi-species (monomer alphabet of size *λ*; polymer of length *ν*) [3], and the feedback arrow is a braid of up to *λ*^2^*^ν^* threads that link each sequence to every member of the entire quasi-species. As the reaction proceeds, and provided the error rate is below a characteristic, system-specific threshold, it drives the self-organising process of Natural Selection in the polymer sequence space. (For polymers of defined length *ν*, the accuracy of monomer copying (the arithmetical inverse (1 − *ε*) of the error rate *ε*) can be regarded as the control parameter for the self-organising transition, due to Natural Selection, from a random population of polymers to a dynamically stable quasi-species.) The polymeric sequence states are often energetically degenerate, or nearly so, and energy differences do not account for Darwinian reproductive success. Rather, it is the kinetics of the monomer condensation process, the copy-error (mutation) rate and the Hamming distance between sequences that account for the “self-organisation of matter”, as Eigen [3] described it.

In a later analysis, Leuthäusser [21] demonstrated a close correspondence of quasi-species selection to a statistical mechanical model: an Ising spin representation of sequences and a Hopfield Hamiltonian representation of rate coefficients. The significance of this analysis is the confirmation that although a supra-physical level of description is required to give a satisfactorily succinct account of Darwinian selection among polymer sequences, the process arises from simple kinetic “interactions” between individual sequences in the absence of any internal or external “guidance”. The absence of guidance in the process of selection is the quintessence of the neo-Darwinian interpretation of evolution [22,23], in terms of which the phenotypic traits that govern the birth and death rates of any genotype are entirely the consequence of material properties and processes. In the case of quasi-species of polymeric sequences, the rates of synthesis and degradation of any individual molecular type in a well-defined solvent milieu could, in principle, be calculated directly from quantum statistical mechanical considerations alone, validating the mechanistic neo-Darwinian outlook. 

Very similar considerations apply to autocatalytic systems with reaction graphs/networks more complex than that depicted in Figure 1, more generally those referred to as “reflexive autocatalytic food-generated (RAF)” sets [24]. Such systems can potentially be guided into different dynamic regimes displaying a range of spatio-temporal behaviours, as demonstrated by the variety of dissipative structural forms displayed by the Belousov—Zhabotinsky system and its relatives [25]. Attempts to build some molecular biological realism into RAF architectures have achieved only limited success [26,27] and are far from finding networks capable of supporting the emergence of coupled polymer replication and translation processes. On the other hand, RAF theory has led to new insights into cellular metabolism and its origins [28,29].

The second fundamental process of molecular biology, ribosomal protein synthesis, the process of translating sequences of trinucleotide codons into sequences of amino acids, is also autocatalytic, although very unlike replication. The autocatalytic character of translation comes from its absolute dependence on the aminoacyl-tRNA synthetase (aaRS) enzymes. Each aaRS attaches a specific amino acid monomer to a “cognate” tRNA molecule that is identified from within the available cellular pool according to the relevant subset of the genetic code’s rules (Figure 1).

Almost every cell exhibits one or more quirky exceptions to the standard genetic code and its attendant aaRS procedures, but neither these exceptions nor the complexities of the overall process of translation are an impediment to teasing out the consequences that the reflexivity of autocatalysis confers on the self-organising dynamics of aaRS [30,31] or ribosomal protein production [32]. We can take the symbol X in Figure 2 to represent a cell’s set of aaRS enzymes, canonically 20 separate proteins, but we would have to add “mRNA, tRNA, ribosome, etc.” to the main reaction arrow as a generic indication of the quasi-catalytic role in protein synthesis of not only genetic information but also the host of other protein and nucleic acid molecules, including ribosomal proteins, initiation factors, etc., and also ribosomal and transfer RNA species. (When focusing on computational aspects of translation, the role of ribosomes, but more especially tRNA, can be sidelined as a complicating factor in the autocatalysis of aaRS synthesis, just as the intermediate formation of a complementary strand does not determine the selective outcome in the case of nucleic acid replication. The role of tRNA in the evolution of coding is discussed in detail by Carter and Wills [33].) In a manner more extravagant than the complex pattern of kinetic feedback seen in quasi-species replication, the synthesis of any single aaRS molecule typically requires the entire suite of cellular aaRS types because nearly every functional protein in a cell includes in its sequence every letter from the amino acid alphabet, or nearly so. This circumstance dictates an extraordinary level of cooperativity in the dynamics of aaRS production and sets stringent limits on the regime of self-organisation within which protein synthesis can be stably maintained. 

The self-organisation of coding as a result of aaRS autocatalysis was first described by Colombano [34], then in more detail by Bedian [30] and again more explicitly by Wills [31,35]; the underlying dynamics of such systems have been laid out by Wills and Carter [36] in Equations (6)–(11) and Figure 2b of that paper. None of this work addresses the evolutionary origin of the mechanism whereby the sequence collinearity of genes and proteins was established. Rather, it addresses the subsequent computational “chicken–egg problem” of how a set of proteins that execute the rules of a code emerged (“Which came first? The proteins or the code?”). First, coding requires a reflexive set of genetic sequences (the “egg”), which, when translated according to the rules of a code, produce a set of protein sequences possessing the aaRS activities (the “chicken”) needed to execute those rules (Figure 3). Second, the proteins (the “chicken”) cannot be produced from the reflexive genetic information unless they are all already present (the mature “hen”) to interpret the information according to the rules of the code. In all known molecular biological systems, the chicken–egg analogy can be taken one step further: the genetic information (essential “egg” component) necessary for protein (“chicken”) production is replicated by protein molecules (“chicken” components), compounding the original dilemma concerning the origin of existentially co-dependent entities. We return to this compound problem in Section 5.

The coexistence of a coding set of aaRS proteins and corresponding genetic information that is reflexively related to the proteins’ amino acid sequences is the sine qua non of aaRS protein-based genetic coding. This is true whether we are discussing its maintenance in every living cell on the planet or its evolutionary origin.

The genetic code operates only because the pattern of a polymer sequence of one type, a nucleic acid, reflects the quantum mechanically determined properties of a particular polymer of another type, a protein, mediated by the alphabet-to-alphabet mapping of the code and the provenance of protein folding. The emergence into reality of this very arbitrary relationship of information pertaining to utterly disparate domains of molecular characteristics, nucleic acid sequences and the catalytic capabilities of proteins is the first imprinting of meaning onto matter, the advent of semantics, of which we have evidence. That the imprinting was apparently the means of its own creation gives cause to reflect on how ordinary physical processes were computationally constrained to produce such an outcome and the consequences attendant upon such a result having been achieved.

## 4. Reflexivity and Self-Organisation


*Thesis: Thermodynamically driven self-organisation provides the means for the mutual population growth of interacting entities, initially produced only transiently, to proceed as far as the codependent species dominating the system. The autocatalytic growth of a coding set of aaRS-like assignment catalysts is self-organising when the production of the proteins is guided by reflexive information. Reflexivity of information arises as an abstract property of an appropriate arrangement of aaRS functionalities in protein sequence space.*


The general practical solution to chicken–egg problems in dynamic systems is self-organisation, whereby there is amplification, through reflexive feedback, of things or circumstances that need one another for their production or occurrence. Self-organisation occurs when an internally cooperative arrangement of co-dependent system elements emerges from local interactions between those lower-level components rather than being driven by mesoscopic or macroscopic external constraints, e.g., thermodynamic parameters. Assume that some mechanism for peptide synthesis was in operation, making random-sequence proteins collinear with RNA sequences by using whatever innate, weak aaRS-like catalytic activity was available within the protein population. Then, suppose, by coincidence, a coding set of assignments momentarily predominated at the same time as reflexive information was available within the population of RNA molecules. This would be a statistical absurdity for an alphabet involving 61 codons and 20 amino acids, but for binary and even quarternary alphabets, it is entirely plausible, as has been demonstrated by computer simulation [31,32]. Such a stochastic event is all that is required to initiate a self-organising process that culminates in a coding set of peptides dominating the entire protein population [32,35,39,40]. 

Once the stochastic fluctuation threshold for coding self-organisation is crossed, the transition from random to coded peptide synthesis is thermodynamically deterministic, driven by the inevitability of the coding set of peptides channelling dissipative resources into their own production. This is the classic trajectory of chemical self-organisation [9]: a threshold of dynamic stability is crossed locally as a result of a statistical thermodynamic fluctuation, and the local composition acts as a seed for the self-organising growth and expansion of the dynamically novel regime. In the case of coding self-organisation of peptide production, the new regime is completely dependent on the presence of the corresponding reflexive genetic information, which can properly be characterised as *guiding* the computational self-organisation in peptide/protein sequence space. In fact, the thermodynamic possibility of chemical self-organisation in the process of peptide synthesis is an artefact of the abstract property of reflexivity conferred on particular sets of genetic information as a result of their nucleotide sequences (see Section 5). Coding self-organisation requires this form of computational guidance. 

From the perspective of quantum mechanics, that is, the laws governing the behaviour of all of the physical particles that comprise a system of RNA-sequence-dependent peptide synthesis, the special property of reflexivity that enables the sequence of a nucleic acid to guide coding self-organisation is occult. It is not, as Küppers [13] would have it, “based exclusively on the known laws of physics and chemistry”. It is a determinative computational condition for coding from strings of nucleotide bases to strings of amino acids, a condition which precedes any realisation under particular quantum mechanical circumstances. Of course, quantum mechanics determines when and where the condition can be met in a quantum mechanical world, but it is not quantum mechanics exclusively that sets off the emergence of genetic coding and its growth to become the dominant computational cause of order throughout the geosphere. Rather, it is the historically contingent occurrence of the condition of reflexivity being instantiated in the available molecular material that sets biology on its path. 

This is quite different from the manner in which some historically contingent, apparently random configuration of matter or energy at a particular place and time acts as a “seed” for the later course of events or formation of structures, e.g., crystal growth, or even replication autocatalysis. In these cases, the chemically functional properties that drive the evolution of the system are quantum mechanically intrinsic to the molecules that have those properties, whereas, in the case of coding self-organisation, the intrinsic functionality of an aaRS molecule contributes to the evolution of the system contingent upon the presence of not only completely different aaRS molecules but also encoding nucleic acid sequences. The molecular biological function of translation depends not only on the intrinsic chemical functionality of molecules but on the historically contingent coincidence of more than one functionality *and* the presence of reflexively encoding information in the form of a nucleic acid with a sequence that is completely arbitrary *vis-à-vis* the quantum mechanical processes occurring in the system.

It is still possible to adopt the philosophical point of view that the only reality is a single cosmic wave function whose deterministic evolution is described by the time-dependent Schrödinger equation [41]. In that case, the existence of genetic coding simply demonstrates that polymers with the necessary reflexive sequence relationships first coexisted accidentally at some time in the distant past, and the event gives the appearance of having triggered the last 4 Gyr of biological evolution. However, it seems more reasonable to view the abstract relationship of reflexivity between polymer sequence patterns as a real cause rather than an apparent cause of coding self-organisation in that its instantiation, whether accidental or engineered, is required at the origin of life or at least the only version of life that we know of so far. If sequence reflexivity is able to guide the self-organising behaviour of a quantum mechanical system, it should be afforded a status equal to the laws of quantum mechanics in respect of the causation of events that have led to the formation of the world we inhabit. This argument is essentially the same as Socrates’ repudiation of mechanistic science 2522 years ago [1]. 

Assembly Theory, recently described in some detail by Marshall et al. [42] and Sharma et al. [43], is likewise predicated on the assumption that although “the laws of physics underpin the origin of life, they cannot predict [its] emergence” and that life can only emerge when “history and causal contingency via selection start to play a prominent role in what exists”. Taking the atomic level as a convenient starting point, Assembly Theory characterises molecules and larger objects in terms of their possible histories of formation. A graph-theoretic quantity called the Assembly Index measures “how much causation was necessary to produce a given ensemble of objects” and has been successfully estimated for molecular systems ranging from single atoms to cells of *E. coli*. The Assembly Index is construed to be an intrinsic property of objects that relates to the possible histories of their appearance in the world. For sufficiently complex objects, their histories may depend on so many past contingencies that it would be impossible to predict any likelihood of their existence from the laws of physics and chemistry alone. Where Assembly Theory differs from what is being proposed here is that it does not posit any necessary role for Schrödinger-type codescript information, let alone reflexivity in the mapping from abstract codescript information to the functional system components, such as enzyme catalysts, that make the assembly of complex biological molecules possible.

## 5. Reflexive Information


*Thesis: Functional reflexivity depends entirely on a set of nucleic sequences specifying, through a code, the positions of the corresponding coding set of AARS-like assignment catalysts in the protein sequence space. The choice of the amino acid sequences of a coding set of aaRS enzymes is completely constrained by how protein functionality varies with sequence, the “functional embedding”. Gene–Replication–Translation models demonstrate the simultaneous, mutually guided self-organisation of both translation and replication as a result of a system-dynamic transition from chemical homogeneity to travelling waves arising from Turing-like reaction–diffusion coupling. This system-wide transition is self-guided.*


The possibility of constructing a reflexive relationship between the sequences of polymers that act as genes, nucleic acids, and those that act as catalysts, proteins, is constrained by the generic properties of these molecules, most especially the distribution of relevant catalytic capabilities within the sequence space of the second category of polymers [31,44]. In the case of proteins, it is clear that the relevant sequence space contains molecules, the aaRS enzymes, with the capability of accurately choosing a unique amino acid from a canonical alphabet of twenty types and, likewise, choosing from a larger pool of tRNA species only those bearing anticodons that correspond to the chosen amino acid according to the genetic code look-up table. The catalytic action of each such aaRS enzyme serves as a logic gate that enforces one of the 20 sets of coding rules for the transfer of information to amino acid sequences from nucleotide sequences. The construction, by reference to genetic information, of proteins able accurately to execute such operations depends entirely on the variation in protein properties, in this case, the specified catalytic capabilities, that can be effected by making amino acid substitutions to move from one point to another in protein sequence space. The many close variants of the standard genetic code, which have remained nearly identical across all known forms of life for the past 4 Gyr, are proof that a suite of ~20 tRNA-aminoacylating logic gate units, albeit of varying molecular sizes, can be constructed using an alphabet of ~20 amino acids spanning a range of sidechain chemical properties, especially as those properties pertain to the nanoscopic chemical environments created when the sidechains are packed in close proximity to one another in a folded protein. However, an information processing system of such complexity could not be the result of a single symmetry-breaking event enabled by a favourable fluctuation in a randomly operating system of RNA-sequence-dependent peptide synthesis. 

In order to address this problem, it is first necessary to clarify how the association between nucleic acid molecules carrying reflexive information and the corresponding set of catalytic functionalities for codon-to-amino acid assignments is achieved. To be maintained, genetic information has to be continually copied and that copying process is inevitably error-prone. Thus, even starting from a “Garden of Eden” state in which all of the genetic sequences are reflexive and all of the protein sequences are exact translations of the genetic sequences, sequence errors introduced by imperfect nucleic acid replication and translation would, at best, reduce the populations of each of the model sequences to a quasi-species distribution of *n*-error mutants (*n* = 0,1,2…). Further accumulation of errors ultimately destroys the genetic “guidance” needed to maintain self-organised coding [35] unless there is a selective advantage favouring coding reflexivity in the available genetic template information. Therein lies an apparently insuperable problem for the hypothesis that coding could arise spontaneously from a system of random RNA-sequence-dependent peptide synthesis. 

Both model simulations [35] and theoretical calculation [2] demonstrate that the random synthesis regime is the only stable dynamic attractor for coupled replication and translation in a “well-stirred” (homogeneous) chemical system. On the other hand, if the composition of the polymer populations, both nucleic acids and proteins, are considered to be functions of the spatial coordinates of the system, it becomes evident that these systems can simultaneously develop and support the maintenance of both reflexive information and its corresponding code [2]. From an analytical point of view, coding is rescued from extinction by the inclusion of diffusion terms corresponding to molecular movement in the coupled chemical rate equations for the production and loss of each molecular species. Starting with a system that is chemically homogeneous, the stable coupling between the replication and translation emerges as a moving wavefront where reflexive information guides coding self-organisation and, vice versa, translation according to local code rules promotes the selection of the reflexive information. The key to this last influence is the catalysis of nucleic acid replication by encoded peptides/proteins. Most evolutionary models of translation consider the existence of such encoded protein replicases of the exact kind found ubiquitously in living systems. Having this feature, the triadic Gene–Replication–Translation (GRT) system (Figure 4) is stabilised along the moving wavefront, the location in space where the production of any one of the three types of molecules is promoted by the presence of the other two.

In GRT systems, translation promotes local gene replication, but errors in either translation or gene replication degrade replicase activity, giving reproductive advantage to nucleic acid molecules at locations where fewer errors occur than in neighbouring regions of space. Because it is the means of producing replicase activity, translation confers a selective advantage on genetic information in any locale where it occurs. This means that self-organisation in the genetic sequence space is *guided* by translation rather than it being determined by the rate constants for nucleic acid replication that are intrinsic to the mechanisms of action of the replicase, which is how quasi-species selection was envisaged in the much more limited system of Eigen [3].

GRT systems represent an extraordinary situation: coding self-organisation is guided by genetic information, which is, in turn, the product of quasi-species selection in the nucleic acid sequence space, and quasi-species selection, the self-organisation of replicating nucleic acid sequences, is guided by translation, the result of cooperative selection in the protein sequence space. Each of these two processes creates a guiding environment for self-organisation in the domain of the other through variation of what amounts to an environmental constraint on the dynamics of processes in that other domain. The mutual guidance of the two self-organising processes, the replication and translation of genetic information, is internal to the system, and, therefore, the historical emergence of genetic coding may be described as an example of self-guided self-organisation. 

This self-guidance of GRT systems depends on the overall state of spatio-temporal self-organisation enabled by the coupling of molecular reactions and diffusion in the physico-chemical domain and the coupling of replication and translation in the computational domain of biopolymeric sequences. It begins with the initial occurrence, then the stable maintenance, of the coincidental grouping of maximally functional molecules together in a small region of space. Coding begins as a historical accident, a marginally improbable coincidence involving molecules whose sequences are related by as yet unestablished functional relationships. These relationships become stablised and are maintained by the self-organising forces of group selection, in this case, the coupled operation of chemical reactions and diffusion. As long as the supply of free energy that drives the nucleotide and amino acid polymerisation reactions does not dry up, the established functional relationships between reflexive genetic information and the corresponding set of coding catalysts for codon-to-amino acid assignments constitute a dynamic “frozen accident” [45] upon which the entire panoply of molecular biology can be built.

The evolution of the code’s amino acid alphabet through a series of steps raises a new series of questions: (i) What was the first step? (ii) How did the dynamic regime in one epoch facilitate the transition to the next? (iii) Why did the amino acid alphabet stop expanding when it did? These questions—and the consequences of our answers to them—are addressed in the next three sections.

## 6. Quasi-Species Bifurcation


*Thesis: The self-organising transition in the dynamics of the most primitive GRT system consists of symmetry-breaking bifurcations in the extant gene and protein populations, starting with the transition from an effectively random distribution of both polymer types to separated pairs of matched quasi-species corresponding to two aaRS-like enzymes that execute a binary code, along with the genes which encode them.*


Random peptide synthesis can be considered as a reference “ground state” of zero information transfer against which any degree of coded information transfer can be measured. In the case that peptide/protein sequences originate through a mechanism that makes them collinear with genetic sequences, random synthesis assigns amino acids to all codons with the same uniform probability distribution. The amino acids are not distinguished from one another in the process of protein synthesis, giving an effective alphabet size of *λ*_eff_ = 1. Likewise, the peptide/protein sequences of length *n* produced from genes through the process of collinear synthesis are indistinguishable from one another. Viewed from the perspective of a “single letter alphabet”, the sequence space of the ground state is a single point that identifies a uniform string of length *n*, with the same letter at every position. Framing the problem in this way makes it clear that a binary (*l*_eff_ = 2) amino acid alphabet {*a*, *b*} would create the simplest possible system of translation. (This excludes non-integral effective alphabet sizes that occur when a single codon has probabilities *p* and 1 − *p* for its assignment to the two amino acids *a* and *b*. This situation can be avoided as long as *p* ≈ 1 for assignment of the codon to *a* and *e* = 1 − *p* ≈ 0 is the probability of an assignment error.) In that regard, neither *a* nor *b* need represent a single amino acid. Rather, they are conceived of as disjoint sets of amino acids whose individual members are chemically distinguishable but functionally indistinguishable *vis-à-vis* the process of translation. (This perspective emphasises the nature of chemical “letters” and their membership in an alphabet. These symbolic entities are operationally defined and do not refer to sets of quantum states of any precisely delineated physical system).

Exactly the same argument applies to the differentiation of codons as they are replicated and translated. Random replication produces a random distribution of genetic sequences, the “null quasi-species”, but ground state translation produces the same random distribution of protein sequences irrespective of the input genetic sequence. Thus, the initiation of translation using a binary code requires the discovery of reflexive information encoding two aaRS catalysts that make exclusive assignments between two distinguishable codons {*A*, *B*} and amino acids {*a*, *b*} consonant with either of the coding structures {*A*→*a*, *B*→*b*} or {*A*→*b*, *B*→*a*}. The null quasi-species, which covers the entire *λ*_eff_ = 2 genetic sequence space uniformly and in which there is a single class of indistinguishable codons, bifurcates into two much narrower, separated quasi-species, each of which is centred on a gene that encodes an aaRS that catalyses one of the two assignments for the code (Figure 5). 

In reality, the first pair of coding assignment catalysts appear to have been the root ancestors of Class I and II aaRS enzymes found in every contemporary living cell, and the genes encoding them were most probably the two complementary, antiparallel sequences of a single double helical nucleic acid [37,38] (see Figure 3). As discussed above, group selection of coding sets of gene and protein sequences can occur and be maintained spatio-temporally by reaction–diffusion processes mediated by an encoded enzymic replicase [2]. 

The process of quasi-species bifurcation takes place simultaneously in both the genetic and protein sequence spaces. Viewed after the event, when there are 2*^n^* points (distinguishable sequences) in each of these binary alphabet spaces, each aaRS quasi-species has narrowed down from a practically uniform distribution covering the whole space, or a large proportion of it, to be concentrated around a pair of points, irrespective of whether we are referring to the gene (codon) or protein (amino acid) sequence space, as illustrated in Figure 3. (The two points in the genetic sequence space are related by antiparallel base pairing in the illustration.) Then, in answer to Question (i), the transition from random peptide synthesis to binary coding is presumed to have been the first step in the evolution of the genetic code. However, the separation of the extant amino acids into two distinguishable sets is unlikely to have been entirely arbitrary, especially when that division had to provide peptides with sequences that folded into reflexively functional structures. This is born out in the historical trace of early peptide structure as it is reflected in the phylogeny of ribosomal proteins [46], the synthesis of which is autocatalytic in a manner similar to that of aaRS [32]. The first step in the transition from broadly non-functional random coil peptides seems to have been a narrowing down into regions of sequence space in which *b*-*b* elements, including *b*-hairpins, were more prevalent [46].

## 7. Digression: Causation in Coding Self-Organisation


*Thesis: The transition from random to coded protein synthesis in GRT-like systems is initiated stochastically as a result of a local fluctuation containing a combination of molecules whose reaction kinetics are beyond the threshold between two regimes of stability in the dynamics of the system. This event potentiates the mutual guidance of two self-organising processes, the replication and translation of genetic information, which together form a self-guided, physically constructive computational system.*


The transition from random to coded protein synthesis is not a result of Darwinian selection. The only identifiable mechanism of evolutionary search that takes place is within the spectrum of spatio-temporal fluctuations that occur due to the changing numbers of individual genes and proteins present within a small locale as a result of the reaction and diffusion of “birth/death” events. The search within the spectrum of fluctuations occurs in parallel at different locations throughout the system, wherein all of the species required to initiate the bifurcation are already being produced at random as a result of molecular turnover. Rather than a laborious Darwinian search consisting of many rounds of genetic mutation and adaptation, that is, the selection of phenotypes with a competitive advantage, the entire event occurs as a rapid saltation of self-organisation, delivering a fully adapted functional unit in a single hit: not only genotype and phenotype but also the mapping between them [14]. Darwinian selection can only be understood in terms of an already well-defined genotype to phenotype mapping or fitness landscape, whereas emergent coding in a GRT-like system maps out its own fitness landscape as it proceeds. It is a self-guided process.

What is the difference between the coincidence of particular genes and peptides in close proximity to one another, an event capable of initiating genetic coding, and the occurrence of all of the accidental mutations that have triggered endless rounds of Darwinian selection to produce and sustain the complexity of the biosphere for 4 Gyr? Both types of historical accident preserve structures that serve as a “record” of the initiating event. They both have consequences that become entrenched in future systems, not just creating opportunities for later developments of a similar kind but also constraining what later developments are possible. For obstinate selectionists such as Dawkins [22] and Dennett [23], accidentalism is the hallmark of biology to the extent that it requires us to dispense with any notion of interaction, influence, meaning or cause that may be construed as transcending physical processes. From this point of view, referred to in the penultimate paragraph of Section 4, the complexities of biological structure and function are derived entirely from the ineluctable laws of physics, ultimately the cosmic wavefunction [41], and random historical happenchance. This is the antithesis of what is espoused in this paper, especially in relation to the characterisation of reflexive genetic information being able to exert a guiding influence on the dynamic trajectory of the self-organising physico-chemical system in which it occurs. The demonstration of Gibson et al. [47] that a denucleated cell can be guided to survive by rejigging its phenotype using the information available in an entirely synthetic genome, an engineered nucleic acid sequence, attests to the potential of genetic information to cause change in a physical system. 

However, is it legitimate to ascribe the influence of the information as emanating from the pattern of the information per se rather than entirely from its physico-chemical mode of instantiation? Surprisingly, Dawkins [48] would seem to give an affirmative answer when he proclaims concerning the wind spreading willow seeds from a tree on the bank of the Oxford Canal: “It is raining instructions out there; it’s raining programs; it’s raining tree-growing, fluff-spreading, algorithms. That is not a metaphor, it is the plain truth. It couldn’t be any plainer if it were raining floppy discs.” Either way, we would still have to regard any information-derived “guidance” derived from a mutation as originating accidentally from an external source and quite different in character from the deliberate human-directed algorithmic guidance of information-processing machines (computers) in a robotic factory. Similarly, the accidental appearance of reflexive information that initiates coding self-organisation is not in itself enough to signify the takeover of causation from simple physico-chemical control. In any case, unless the replication of the necessary genetic information has a circumstantial selective advantage, its existence is temporary, and any trace of it is eventually lost to mutation [35]. Only if the selective advantage is defined by the information’s “correct” expression can it survive.

Additionally, that is the point. In GRT-like nucleoprotein systems, the reflexive information needed for coding self-organisation is maintained by the *internally guided selection* that hitchhikes on the back of the physico-chemical process of coupled reaction–diffusion self-organisation, or some other form of the dissipatively sustained spatio-temporal ordering of processes in interacting compartments. Reciprocally, the ordered dissipative structure is able to propagate itself only if the reflexive information is there to order the polymerisation reactions to produce protein sequences corresponding to the catalysts needed for translation. In other words, the ordered physical structure required for coding self-organisation in GRT-like systems would never occur were it not for the arbitrary computational relationship of reflexivity between genetic information and the sequences of aaRS catalysts. This computational relationship and its attendant processes of translation that operate on string information, albeit information instantiated as polymeric sequences, enable, constrain and control the basic physical structures that underpin all of biology. The molecular composition and self-organised structure of GRT systems, which include all living cells, are the result of physically constructive molecular computation. 

The advent of integrated computational causation over and above the inexorable trajectory of quantum mechanical events marked a local transition that introduced into the cosmos entities of a sort completely different from anything that had previously existed, not even hinted at by the laws of nature that had evident effect until then. The major symmetry-breaking transitions that had occurred previously and which initiated new epochs of cosmic history did not require the arrangement of particles of matter into configurations that funnelled dissipative resources into self-referential computation, but the advent of the genetic code *did*, with consequences and the creation of modes of existence, eventually including intelligence, vastly different from anything previous.

## 8. Semantic Code Development


*Thesis: Each code-expanding quasi-species bifurcation potentiates the next, enabling stepwise refinements in the genetic specification of protein encoding. Such specification is mediated by the controlled deportment of amino acids with diverse physico-chemical properties to produce folded proteins with increasingly refined functionalities. The enormous combinatorial complexity underlying the interactions between the chain of amino acids comprising a protein imbues the folding process with an evolutionary learning capacity akin to that of the neural networks employed in systems of artificial intelligence. The evolution of the aaRS enzymes records the reflexive harnessing of diverse amino acid properties to construct an information-based, coded operational map of those very physico-chemical properties of amino acids in terms of nucleotide triplet codons.*


It is a very long way from an error-prone binary coding system operated by a handful of nucleic acid and protein species to the technologically dominated geosphere of today. A binary code, which divides codons and amino acids into only two classes, has meagre means of controlling with much specificity the chemical interactions of multiple, disparate nucleotides and amino acids, or for that matter, any other class of organic molecules [49,50]. It is little wonder then that fully established molecular biology has relied on the much richer genetic code, which generically maps 61 trinucleotide codons onto 20 amino acids, since anything recognisable as an archaeal or bacterial cell had come into existence. The question to be answered is how that development took place. The phylogeny of aaRS enzymes gives a clearer picture [51] without revealing what drove the development. Purely Darwinian models can be built in terms of gene duplication and subsequent selection [52,53], but the mechanism of quasi-species bifurcation offers an explanation based on reciprocal links between the physical and computational aspects of the process.

Consider the expansion from a binary code {*A*→*a*, *B*→*b*} to a ternary code {*A*→*a*, *B′*→*b′*, *C*→*c*} based on the splitting of the *B* codon and *b* amino acid classes into subclasses {*B′*, *C*} and {*b′*, *c*}. Because these ternary subclasses are indistinguishable by the binary coding processes, the binary-alphabet aaRS gene and protein quasi-species will occupy the multiplicity of points that include all of the random *B′*/*C* and *b′*/*c* alternatives in the ternary-alphabet sequence spaces arising from occurrences of *B* and *b* in the relevant binary-alphabet sequence spaces. This means that the binary coding system is ripe for a takeover by individual ternary-alphabet molecular species that optimally recognise the ternary-alphabet distinctions that are invisible to the binary-alphabet quasi-species in which *B′*/*C* and *b′*/*c* alternatives occur randomly. The process mimics exactly the initial bifurcation whereby binary coding can emerge from random gene and protein synthesis with the additional feature that the expansion of the alphabet (*λ* = 2→3) facilitates additional increases in not only the specificity with which the aaRS catalyst for the *A*→*a* assignment can recognise its substrates, *A* and *a*, but also, in all probability, the speed at which it can catalyse the assignment reaction. In that case, this second step of coding self-organisation could be viewed as achieving much more than the first step, which placed it within reach.

The initial establishment of a binary *λ*_eff_ = 2 code prepares the system for expansion to a *λ*_eff_ = 3 ternary code by focusing protein production on quasi-species which contain the aaRS variants that are able separately to recognise the amino acid and codon variants, {*b′*, *c*} and {*B′*, *C*}, respectively (or alternatively {*a′*, *c*} and {*A′*, *C*}, leaving *B* and *b* intact) whereby the expanded code is defined. The initial self-guidance of coding self-organisation facilitates a further transition of the same character, emphasising the manner in which the development of information processing at the origin of life emancipates itself from the underlying, ongoing, quantum mechanical causation that governs physico-chemical aspects of the process. As with the initial transition from random peptide synthesis to the operation of a binary code, there is a stochastic “waiting time” for the dynamic threshold of the *λ*_eff_ = 2→3 transition to be crossed, and the waiting time depends on spatio-temporal distribution of the computational events that coincidentally produce aaRS molecules with *B′*→*b′* (or *A′*→*a′*) and *C*→*c* assignment specificities. However, once again, it is the actual satisfaction of the computational condition to which causation of the transition should be ascribed, not just the relentless course of stochastic, nanoscopic, quantum mechanisms. 

Then, in answer to Question (ii), this argument can be followed through every expansion in the effective amino acid and codon alphabets (*λ*→*λ* + 1) all the way to the mature versions of the genetic code that have apparently existed since the time of the Last Universal Common Ancestor of all organisms, or the approximate point in evolution that such a theoretical construction represents. That does not imply every expansion step need be attributed to the mechanism of bifurcation of non-compartmented spatially distributed quasi-species. Variations in the coding table from organism to organism or organelle to organelle attest to the likelihood of the most recent changes having occurred according to more conventional Darwinian mechanisms of code expansion, especially in its later stages [51,54]. However, the explanations offered by the possibilities and speed of self-organised coding development with polymers in their current molecular biological roles—genetic information storage using nucleic acids, aaRS coding assignment and nucleic acid replication using proteins—are far more parsimonious than the alternative RNA World model, which eventually requires a complete switch in both coding hardware and software without loss of “learning” in relation to the postulated adaptation of RNA catalysis to vital system tasks. 

This bears on another largely overlooked aspect of the way in which genetic coding is embedded in molecular processes. An aaRS assignment catalyst is required to differentiate accurately between potential amino acid substrates and accept only those that are “correct” according to code rules for the current alphabet size. Proteinaceous aaRS enzymes are created by concatenating amino acids together in genetically specified sequences, but the covalent peptidyl structure of a protein is not, of itself, sufficient to render it functional. Functionality is only achieved when the peptidyl chain is folded within its physiological milieu: an aqueous environment with pH, ionic strength, temperature, pressure and other parameters, sometimes the concentrations of certain ions, within certain ranges. Under such circumstances, the protein will form its folded functional structure, a glimpse of which can be achieved by using a variety of experimental techniques such as X-ray crystallography, nuclear magnetic resonance and cryogenic electron microscopy. These techniques, now supplemented by artificial intelligence algorithms, reveal which of a protein’s amino acid residues lie in the neighbourhood of one another and the extent and mode of their interactions.

The mapping from genetic information onto the molecular biological function—its phenotypic expression—is *semantic*. In the processes of inheritance, genetic information acts as a digital proxy representation of essential functionalities. In the case of the catalytic functions of enzymes, the first step in giving meaning to the “written” genetic description is the translation from the codon alphabet of nucleic acids into a chain of letters chosen from the amino acid alphabet of proteins by application of the code’s mapping. The peptide chain then folds so that its individual residues are positioned in a pattern that creates the function of the enzyme, a subset of that protein’s phenotypic properties, dependent on the spatial and temporal disposition of the amino acids, their diverse chemical properties and the detailed interactions between them. However, that is not something at all simple, depending as it does on the complex quantum chemistry of amino acid sidechains and their interactions. Modes of the protein’s intramolecular mobility enable allosteric effects to propagate, sometimes entailing uniquely quantum phenomena, from one location to another through successive “onion skin” layers of surrounding amino acid sidechains. Furthermore, such effects may be part of the stochastically canalised series of events which constitute the reaction steps of enzyme catalysis. Altogether, the mapping from the amino acid sequence of a protein onto the values of the kinetic parameters that determine catalytic functionality is immensely complex. It has proved impossible to describe the mapping in terms of a transparent algorithm or set of rules. On the other hand, the protein-folding problem has yielded to systems of machine learning [55] in which modelling is surrendered to the pragmatics of adjusting the strengths of combinatorial interactions between entities that have no explicit identification with system elements or features.

Included in the range of biochemical reactions whose systemic control depends on protein folding is the set of codon-to-amino acid assignments made, according to their inherent specificities, by the population of aaRS enzymes present in the system. This means that the very symbols, the codon and amino acid alphabets, in terms of which the code can be expressed, are defined reflexively. The symbol-recognition operations that define the effective codon and amino acid alphabets are executed by aaRS enzymes, whose overall capability to do so is constrained by the net specificity that can be achieved by folding, into functional spatial arrangements, sequences of individual amino acid residues, every one of which, in each instance, is chosen with the limited level of precision afforded by the current alphabet. The semantic mapping whereby genetic information is given functional meaning is self-constructed: the material instruments of meaning-making are constructed by application of the meaning-making functions available in the form of those instruments themselves. Kauffman and Lehman [56] give detailed consideration to the origin of the semiotic (alphabet-to-alphabet) mapping of the genetic code, but they do not go on to address what role feedback in the semantic (information to function) mapping may have played in determining the structure of the code.

The reflexive “chicken–egg” feedback in the process of defining an alphabet based on the differing chemical properties of amino acids signals the possibility of the code evolving through self-organising saltations from one codon-to-amino acid mapping to another of higher alphabet dimension. Then, in answer to Question (iii), the expansion of the effective size of both amino acid and codon alphabets would be expected to continue until no functionally significant gain in the specificity with which the letters of the alphabet can be differentiated is to be gained by a further expansion step. Compared with the time period for which the genetic code has remained quite stable in its extant state, its development after the Earth’s crust cooled enough to make aqueous chemistry possible must have been remarkably rapid. Additionally, what is more, we should expect a feedback process enabling a code to evolve so rapidly to produce one that is not only a mapping from nucleic acid to amino acid sequences but also a map that reflects, in the space of trinucleotide codons, the underlying chemical properties of amino acids, especially those properties that either mediate the folding of protein chains and or are relevant to specific catalytic function. This characteristic of the genetic code has been clearly demonstrated by Carter and Wolfenden [57].

There is an evident connection between the reflexive, self-organising construction of the amino acid alphabet and the mapping of the physico-chemical properties of amino acids onto other symbols (codons), whose main role is simple data storage. This connection is profound. The diverse properties of amino acids are the elements of chemical behaviour put to use in subjecting basic molecular biological processes to computational control. Additionally, the putting to use of those differential properties is itself a computational process. It is effected not by direct chemical influence but by making choices between letters from a quite different alphabet, the low-dimensional symbol space of trinucleotide codons. This marks the emergence of genetic coding as the first local manifestation of representation and meaning—immaterial semantic aspects of reality—coming to have a determinative influence in the local cosmos. Through the structural regularities of the code certain generic physico-chemical properties of amino acids are embedded in the bitwise differentiation of nucleotide triplets [33]; this, in turn, optimises the adaptation of the code for the construction of proteins able to produce and control quantum mechanical nano-environments in which individual covalent bonds can be formed and broken selectively and specifically. 

This is like having an algorithmic rule set that has been written to facilitate the writing of programmes for their efficient implementation on a particular hardware platform. The difference is that in this case the hardware platform is the world of aqueous chemistry and the rule set has emerged through a process of self-guided self-organisation rather than having been created by some external intelligence, either natural or artificial. Proteins are able to act as switching devices that decide on events within regions of space as small as a hundredth of a cubic nanometre, depending on whether some arbitrary condition of structure or interaction is met at a relatively distant location. A classic example is the aminoacylation with a specific amino acid of the 3′-terminal CCA acceptor stem of a tRNA by an aaRS enzyme, depending on the presence of a cognate anticodon at a site that is 7.5 nm remote, at the opposite end of the tRNA “clover-leaf” structure.

It seems doubtful that anything resembling biology in the water-dominated environment on this planet could have occurred without the initial emergence of a system of reflexive information processing similar to that of the genetic code. In the same way as Schrödinger [6] recognised the need for a nanoscopic system of information storage in order for the inheritance of mammalian characteristics such as the Habsburger Lippe to transverse, generation after generation, the single cell stage of a human zygote, so too might we hypothesize the apparent impossibility of precisely controlling, coordinating and speeding up all of the reactions and other processes inside a Mycoplasma cell as small as 200 nm in diameter without some form of quasi-algorithmic, hardware-adapted information processing, such as the universal molecular biological system of genetic coding.

## 9. Concluding Remarks

A considerable proportion of the bulk material in the geosphere is comprised of constituent molecules and arrangements of them, which possess structural complexity of magnitude hyper-astronomical compared with that of any other material so far found in the known cosmos. If we could convert all of the matter in the universe into randomly synthesised proteins of typical size, there would be a near-zero probability that any two such molecules would have the same atomic structure. Yet, in biomass such as a large forest, a limited range of protein molecules each occurs in copy numbers of the order of Avogadro’s number. The origin of any planetary-scale bulk of material displaying such extraordinary order and molecular complexity poses a scientific question akin to, and as significant as, the origin and structure of galaxies, solar systems, and planets, as well as their geology. The questions of astronomy and cosmology are best answered in terms of the operation of what counts as the four fundamental forces of physics and the laws of thermodynamics. However, there is no straightforward explanation of the order and complexity of biological material in such terms. 

The biosphere is not the simple outcome of matter being shaped and transformed by mechanical, electromagnetic and chemical forces under the influence of gravity and planetary transfers of radiation. Rather, its molecular composition is the result of the operation of a multitude of control networks constraining and guiding molecular processes on timescales ranging from femtoseconds to Gigayears and over length-scales ranging from nanometres to Megametres. The fundamental forces of physics manifest effects over even wider ranges of time and length scales, so the ranges of scale typical of biological processes are not a special feature of biomatter which distinguishes its production and composition from matter of inanimate origin. Nor does physico-chemical analysis bring to light any special quantum mechanical feature of molecular components that are produced by biological processes compared with those emanating from other natural or artificial processes. Crude vitalism has been judged on the evidence to be scientifically fallacious. 

As empirical investigations into biomatter have progressed over the last 150 years, it has become clear that its production depends on the continual storage, copying and transformation of polymer sequence information. Genetic information in organisms plays a role akin to computational bits, the Boolean values of which are registered to allow physical changes to be controlled. An organism’s genome contains a library of information relevant to its current structure and workings, some of it having first appeared in nucleic acid sequence records which date back to the beginning of biology and have constrained processes within every living cell that has existed in the entire interim period to the present. In fact, every bit of genetic information has a phylogenetic history or, more especially, an origin in some contingent historical event related to the disposition of interacting molecules and the pathways available for free energy dissipation or whatever else drove the selection of that bit of information or its creation in some other self-organising process. 

Discussion of the “Life = matter + information” formula [11,12] usually focuses on information whose quantity can be measured entropically [7] and whose processing can be observed as a result of various molecular mechanisms. However, if the formula were transformed into ‘Life = matter + computation”, it would give a much clearer picture of what happens in cells: their molecular composition can only be understood as the result of information *processing* over and beyond the mundane making and breaking of chemical bonds in accordance with the probabilities of quantum mechanical outcomes and inevitable overall increase in thermal entropy. By any standard of observation, it is the input of genetic information into a reflexively constructed code-based computation that is responsible for functional proteins being found in cells, not some much finer-grained quantum ordering effect that is yet to be discovered. The flows of energy and molecular material that occur in cells are coordinated and regulated in a richly intertwined network of molecular control processes that subjugate the laws of physics and chemistry to the role of mere “substance effects” rather than “causes”, as, for example, Socrates [1] would have used the term.

It is commonly asserted that all biological phenomena depend exclusively on the rules of quantum mechanics and can be explained without reference to the computational relationship of reflexivity between gene and protein sequences needed for coding. This seems similar to propounding that until the first demonstration of stellar parallax by Bessel in 1838, further elaboration of the Ptolemaic geocentric construction was all that was really required to explain astrometric observations of planetary motion. That would virtually annul the extraordinary contributions of Copernicus, Kepler, Galileo and Newton to the interpretation of astronomical measurements over the three centuries prior to Bessel’s measurement and would be, in Socrates’ terms, “an extremely slipshod way of thinking”. 

## Figures and Tables

**Figure 1 entropy-25-01281-f001:**
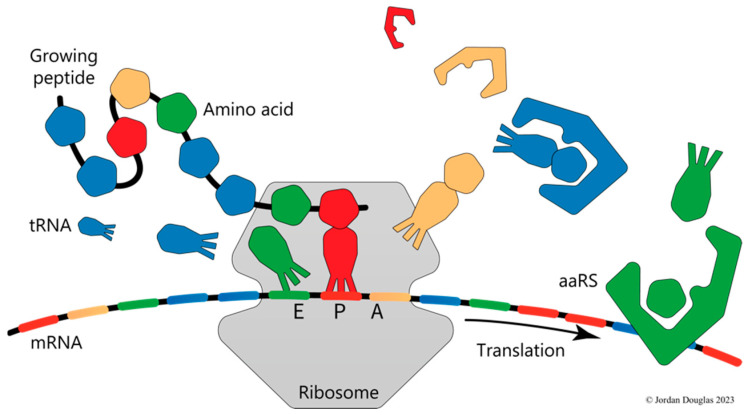
Ribosomal protein synthesis. This representation of protein synthesis emphasises the computational role of aminoacyl-tRNA synthetase (aaRS) enzymes in translation. These enzymes (irregular shape, RHS of figure) enforce the rules of the code by specifically matching their amino acid substrate to cognate tRNA molecules, i.e., those bearing nucleotide triplet anticodons (drawn as “legs”) consistent with the codon-to-amino acid rules of the genetic code, shown here as colour-matching. A 4-letter code (different colours for different letter symbols) is depicted. The amino acid added to the growing peptide is a correct (colour-matched) translation of the codon occurring at that point in the genetic message (mRNA).

**Figure 2 entropy-25-01281-f002:**
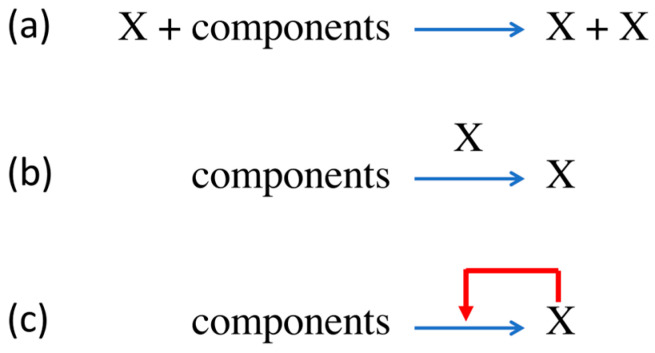
Replication as autocatalysis. The net reaction, components → X, is catalysed by X, a role shown by (**a**) adding it to both the reactant and product side of the chemical equation; (**b**) placing it above the arrow; or (**c**) indicating catalytic feedback (red arrow).

**Figure 3 entropy-25-01281-f003:**
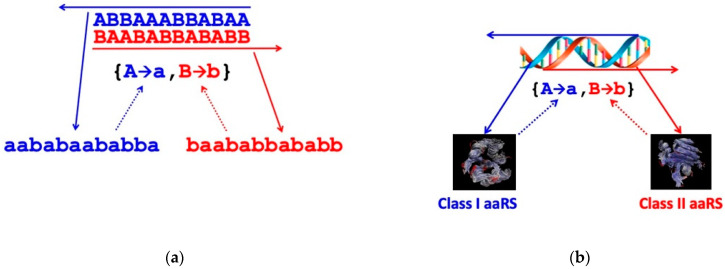
Reflexive production of assignment catalysts. (**a**) Two fictional aaRS-encoding genes, AABABAABABBA and BAABABBABABB, are shown as anti-parallel complementary strands of a single base-paired nucleic acid molecule in accordance with the hypothesis of Rodin & Ohno [37,38]. The genes are translated by operation of the coding assignments A→a and B→b between the binary codon alphabet {A, B} and the binary amino acid alphabet {a, b} to produce molecules with amino acid sequences aababaababba and baababbababb representative of two separate classes of aaRS species. (**b**) The antiparallel genes are depicted in a double helical configuration, and the conserved core structures of the ancestral Class I and II aaRS enzymes are shown.

**Figure 4 entropy-25-01281-f004:**
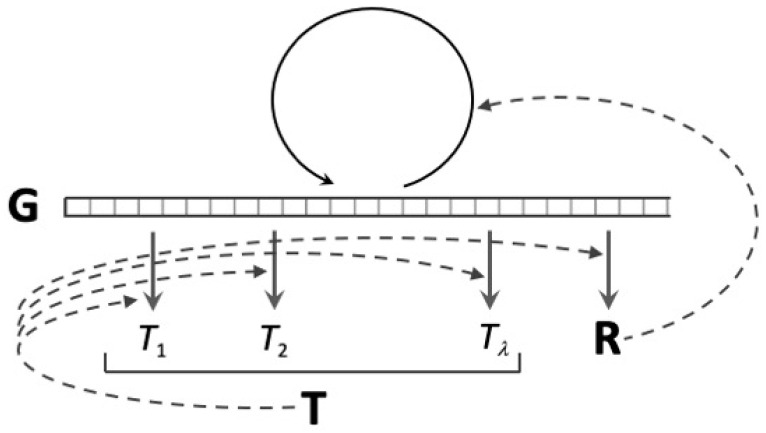
Gene–Replication–Translation (GRT) system. The genetic information in the nucleic acid **G** encodes a set of aaRS-type assignment catalysts **T** = {*T*_1_, *T*_2_ … *T*_λ_} and a nucleic acid replicase **R**. The set **T** catalyses the codon-to-amino acid assignments for the system’s genetic code.

**Figure 5 entropy-25-01281-f005:**
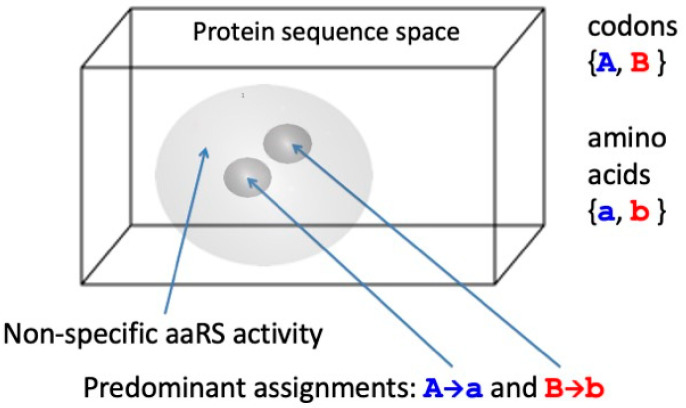
Quasi-species bifurcation. Proteins with aaRS-like activity, including weak and non-specific activity, are found throughout a large region of *n*-dimensional protein sequence space. However, significant activities for the two specific assignments, *A*→*a* and *B*→*b*, are found within much narrower, separate domains, which are also separate from the domains of the code conflicting *A*→*b* and *B*→*a* assignments (not shown). Binary coding arises from a symmetry-breaking transition whereby the protein population with aaRS-like activity becomes concentrated from the large, broad domain into the two smaller regions of sequence space.

## Data Availability

No new data were created in this study.

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
