# Peer review of "Origins of Genetic Coding: Self-Guided Molecular Self-Organisation"

_entropy, 2023, doi:10.3390/e25091281_

Round 1

Reviewer 1 Report

The article reads like a philosophical treatise without precisely formulated theses to back it up. The text is beautifully written. However, it remains unclear what the new and original ideas and/or key conclusions are. It is therefore only logical that the section "Discussion and conclusion" in particular is very vaguely formulated. The few concrete assertions, such as "In fact, every bit of genetic information has a history or, more especially, an origin in some contingent historical event related to the disposition of interacting molecules and the pathways available for free energy dissipation", remain without evidence. Therefore, I suggest the author to rethink the structure of the article and to formulate his theses more sharply and precisely.

Reviewer 2 Report

Journal: Entropy

Manuscript number: 2525019

Title: Origins of genetic coding: self-guided molecular self-organisation

In this work several concepts on the origin and nature of the genetic code are put forward. The computational character of the genetic code is emphasized. Its emergence in a pre-biotic world is envisaged by the incidental satisfaction of a condition of reflexivity between polymer sequence information and system elements able to facilitate their own production through translation to that information. This event is depicted in the dynamics of Gene-Replication-Translation systems, as a process of self-guided self-organisation. The spontaneous emergence of a primordial genetic code between two-letter alphabets of nucleotide triplets and amino acids is easily possible starting with random peptide synthesis that is RNA-sequence dependent. The evident self-organising mechanism is the simultaneous quasi-species bifurcation of the populations of information carrying genes and enzymes with aminoacyl-tRNA synthetase-like activities. The origin of genetic coding is characterised as an event of cosmic significance in which quantum mechanical causation was transcended by constructive computation. This mechanism allowed the code to evolve very rapidly to the ~20 amino acid limit apparent for the reflexive differentiation 20 of amino acid properties using protein catalysts. The self-organisation of semantics in this domain of physical chemistry conferred on emergent molecular biology exquisite computational control over the nanoscopic events needed for its self-construction. The transition took place in a domain of activity governed not by quantum mechanics and general relativity but by the formation and processing of referential information. In that domain chemical groups took on an immaterial, abstract status, that of digital symbols, a transition that was made possible by reflexivity, a pattern of self-referential information that arose as a side effect of an autocatalytic chemical cycle: the information in some extant patterns of symbols was processed computationally to produce the molecular machinery needed to execute the steps of that very computation.

MAJOR

1. The goal of the manuscript must clearly be stated. The assumptions/limits of each claim or concept must be added.

2. tRNA molecules are not mentioned in the manuscript. Yet they played a pivotal role in the formation of the genetic code. It is generally accepted that mini-helices of proto-tRNAs were capable of self-aminoacylation prior to the emergence of the earliest synthetases, a process proposed to underlie the formation of the genetic code. Two codes appeared in tRNA molecules: the operational and the anticodon code refined by identity elements. How do these facts change/reinforce the appearance of reflexivity of the genetic code?

3. The genetic code is considered as a Turing machine. To describe or to define what a Turing machine is would be beneficial to a wide audience. A Turing machine is the combination of a sequential, finite-state machine plus an external read/write memory storage medium called the tape. The tape is a linear sequence of squares/spaces, with each square holding one of several possible symbols. More generally, a Turing machine can have any number of different symbols it can recognize. As far as I know, nobody has ever actually constructed a Turing machine. It is a purely theoretical concept. Given enough time, a Turing machine can compute anything that can be computed. A Turing machine’s power to compute comes not from super technology, but from its tape, for 2 reasons. First, Turing was the first to conceive the idea of a stored program that could be changed by the operation of the machine itself, A Turing machine has the ability to remember what has happened in the arbitrarily distant past.   

4. I suggest pinpointing the A, P, and E sites in Figure 2.

5. In Figure 3, two fictional aaRS-encoding genes are depicted as anti-parallel genes in double-helical structure. I suggest using anti-parallel strands since the double helical structure of DNA appeared after LUCA. The first genes were made of RNA.

6. This work does not support the mutation-selection balance of long polymers. This model posits the advent of Darwinian evolution before the appearance of protocells. This is the case with Eigen’s theory of template-based replication of genetic polymers (Eigen, 1971; Eigen and Schuster, 1977; Eigen et al., 1988).

Eigen, M. (1971). Self-organization of matter and the evolution of biological macromolecules. Naturwissenschaften 58, 465–523.

Eigen, M., McCaskill, J., and Schuster, P. (1988). Molecular quasi-species. J. Phys. Chem. 92, 6881–6891.

Eigen, M., and Schuster, P. (1977). The hypercycle. A principle of natural self-organization. Part A: emergence of the hypercycle. Naturwissenschaften 64, 541–565.

7. A threshold can be defined as a major qualitative change undergone by a physical-chemical system upon relatively minor changes in the values of systemic or environmental control parameters. One of the most studied thresholds is Eigen’s ‘‘error threshold” which constrains replication-based scenarios of early genetic polymers (Eigen, 1971).

Which was the control parameter that drove the bifurcation?

How was the computer program affected by this threshold?

What does run the genetic program? Dissipative resources or/and reflexive genetic information?

Round 2

Reviewer 1 Report

The author has taken into account the reviewer's comments. A clear formulation of his theses has contributed a lot to understanding his point of view and the comprehensibility of the article.

Reviewer 2 Report

The author answered almost all my concerns. The presentation improved considerably.